# Design and realization of topological Dirac fermions on a triangular lattice

Maximilian Bauernfeind[1,2,7], Jonas Erhardt[1,2,7], Philipp Eck[2,3,7], Pardeep K. Thakur[4], Judith Gabel[4], Tien-Lin Lee[4], Jörg Schäfer[1,2], Simon Moser[1,2], Domenico Di Sante[3,5,6], Ralph Claessen[1,2✉] & Giorgio Sangiovanni[2,3✉]

Large-gap quantum spin Hall insulators are promising materials for room-temperature applications based on Dirac fermions. Key to engineer the topologically non-trivial band ordering and sizable band gaps is strong spin-orbit interaction. Following Kane and Mele's original suggestion, one approach is to synthesize monolayers of heavy atoms with honeycomb coordination accommodated on templates with hexagonal symmetry. Yet, in the majority of cases, this recipe leads to triangular lattices, typically hosting metals or trivial insulators. Here, we conceive and realize "indenene", a triangular monolayer of indium on SiC exhibiting non-trivial valley physics driven by local spin-orbit coupling, which prevails over inversion-symmetry breaking terms. By means of tunneling microscopy of the 2D bulk we identify the quantum spin Hall phase of this triangular lattice and unveil how a hidden honeycomb connectivity emerges from interference patterns in Bloch $p_x \pm ip_y$-derived wave functions.

[1] Physikalisches Institut, Universität Würzburg, Würzburg, Germany. [2] Würzburg-Dresden Cluster of Excellence ct.qmat, Universität Würzburg, Würzburg, Germany. [3] Present address: Institut für Theoretische Physik und Astrophysik, Universität Würzburg, Würzburg, Germany. [4] Present address: Diamond Light Source, Harwell Science and Innovation Campus, Didcot, UK. [5] Present address: Department of Physics and Astronomy, University of Bologna, Bologna, Italy. [6] Present address: Center for Computational Quantum Physics, Flatiron Institute, New York, NY, USA. [7] These authors contributed equally: Maximilian Bauernfeind, Jonas Erhardt, Philipp Eck. ✉email: claessen@physik.uni-wuerzburg.de; sangiovanni@physik.uni-wuerzburg.de

The electronic wave functions of quantum spin Hall materials wind in momentum space in a topologically distinct way from ordinary insulators, as described by the corresponding $\mathbb{Z}_2$-invariant. The quantized transport via spin-polarized boundary modes is protected by time-reversal symmetry, making quantum spin Hall insulator (QSHIs) technologically attractive[1,2]. Ideal platforms are two-dimensional (2D) honeycomb systems, as these naturally host massive Dirac fermions at the $K/K'$ points in momentum space. As drawn in Fig. 1a, spin-orbit coupling (SOC) opens a non-trivial gap, whereas inversion-symmetry breaking (ISB) counteracts SOC and favors the trivial phase[3]. To achieve room-temperature operability, a large gap is essential. Graphene is, for example, a poor QSHI because its SOC arises from weak 2nd-nearest-neighbor hopping processes between out-of-plane $p_z$ orbitals[4]. To improve on the gap size, 2D materials made of heavier elements relying on local SOC are hence superior[5–9]. So far, this type of materials design has been achieved only in bismuthene, a honeycomb system featuring purely planar bonding of its $6p$ orbitals[10].

The strategy of replacing carbon with heavier atoms faces two serious challenges. First, in the unavoidable presence of a substrate high-$Z$ elements tend to form buckled structures that are hostile to topology[11]. Second, many elements order preferentially in triangular rather than in honeycomb lattices when deposited on hexagonal templates (Supplementary Note 1), with no experimental realization of a QSHI phase hitherto[12]. Synthesizing triangular QSHI would therefore potentially accelerate the steps towards the first single-layer QSHI device concept as, for instance, the growth process would profit from the simplicity of a non-bipartite lattice.

Here, we realize a triangular lattice of indium on SiC(0001) with topological band inversion at the valley momenta $K/K'$. In "indenene", SOC arises locally from In $5p$ orbitals and opens a gap between valence and conduction bands of about 100 meV. A global gap is guaranteed by the presence of the substrate which induces an anticrossing of the bands derived from the indium $p_z$ orbital and the two planar $p_\pm \propto p_x \pm ip_y$ chiral orbitals, respectively[13]. The valence and conduction bands are further spin-split at the valleys, as a consequence of the in-plane ISB of SiC(0001). The C atom of the surface SiC layer renders the two halves of the indenene unit cell (labeled as "A" and "B" in Fig. 1b and hereafter) inequivalent.

The splitting of the Dirac bands at $K/K'$ allows to determine the topological nature via a direct energy-resolved analysis of the bulk bands: As illustrated in Fig. 1c, the phases of the $p_\pm$-derived Bloch wave functions give rise to constructive and destructive interference in A and B. The resulting charge localization at the voids of the triangular lattice induces an emergent honeycomb connectivity (see cyan-green and orange spots in Fig. 1b). The energy ordering of these valley-states and the A/B character of the corresponding bulk wave function distinguishes the QSHI from the trivial phase in an unambiguous way, as sketched in Fig. 1a. This, therefore, stands out as an example of a topological classification through the spatial symmetry of the electronic bulk wave functions probed by scanning tunneling spectroscopy (STS).

## Results and discussion

**Theoretical model.** For a microscopic understanding of the physics of a triangular model of $p$ orbitals on a substrate we use the three spherical harmonics

$$\left\{ p_\pm = \frac{1}{\sqrt{2}}(p_x \pm ip_y),\ p_z \right\}. \tag{1}$$

In this basis, we step-wise introduce the key interactions relevant to stabilize the QSHI phase with low-energy Dirac states at $K/K'$, as illustrated in Fig. 2. The following tight-binding Hamiltonian captures the low-energy electronic structure of any realistic implementation, as we will see later in the density functional theory (DFT) calculations for indenene on SiC(0001). The latter is the actual material realization that we propose here and our modeling allows us to precisely determine the conditions under which its QSHI phase is realized.

In the freestanding triangular layer (Fig. 2a, d) the $D_{6h}$ point symmetry yields Dirac-crossings of the $p_\pm$ in-plane orbitals at $K/K'$ and prohibits the hybridization with the $p_z$ subspace resulting in a metallic phase. Local (atomic) SOC ($H^{SOC} = \lambda_{SOC}\mathbf{L}\cdot\mathbf{S}$) becomes relevant at band touchings and opens a gap at $K/K'$ between the $J = 3/2$ and $1/2$ states of size $\lambda_{SOC}$ within the in-plane subspace (see inset to Fig. 2a).

In the presence of a homogeneous substrate (Fig. 2b, e), the mirror symmetry along the surface normal direction is broken and a hybridization gap opens between the in- and out-of-plane orbitals. Considering SOC, a non-trivial insulating ground state is realized and the spin-degeneracy of the states with mixed $p_\pm$ and $p_z$ orbital composition is lifted. This Rashba-like splitting does not involve the Kramers doublet of the low-energy states at $K/K'$ that are protected by the $C_{6v}$ symmetry (see inset to Fig. 2b).

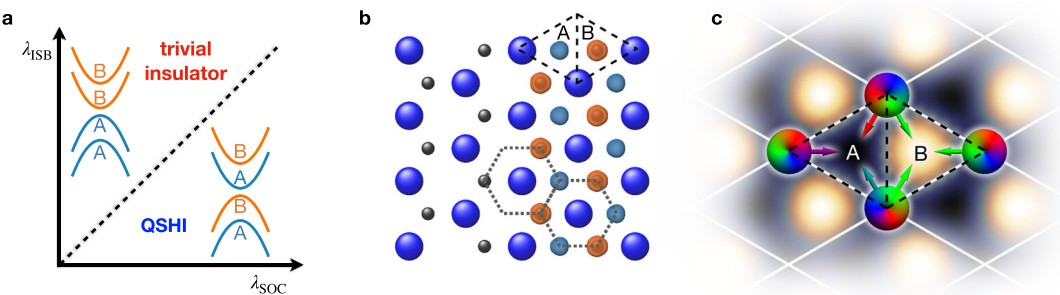

**Fig. 1 Dirac fermions on a triangular lattice. a** Sketch of the phase diagram as a function of SOC ($\lambda_{SOC}$) and ISB ($\lambda_{ISB}$). The insets illustrate the band ordering of the Dirac fermions at the $K$ and $K'$ points in the QSHI and trivial insulator phase. The coloring and the A/B labels correspond to the charge localization in **b**. **b** Triangular lattice (blue spheres) with substrate-induced ISB, as indicated by the C atoms (black spheres) in the first bilayer of the SiC(0001) substrate (Fig. 3a, c). The right half shows a honeycomb lattice (dotted gray lines) emerging from the charge localization of the chiral Dirac states, which peaks in the voids of the triangular lattice. **c** Interference mechanism between Bloch and orbital phases determining the charge localization, shown schematically for the example of a $p_-$ orbital at the $K$-point: the total phases (indicated by the rainbow color scheme) of neighboring sites contributing to the Bloch wave function interfere constructively at the B site. This promotes a high (bright) charge density centered around B, while destructive interference suppresses the charge localization in the A triangle.

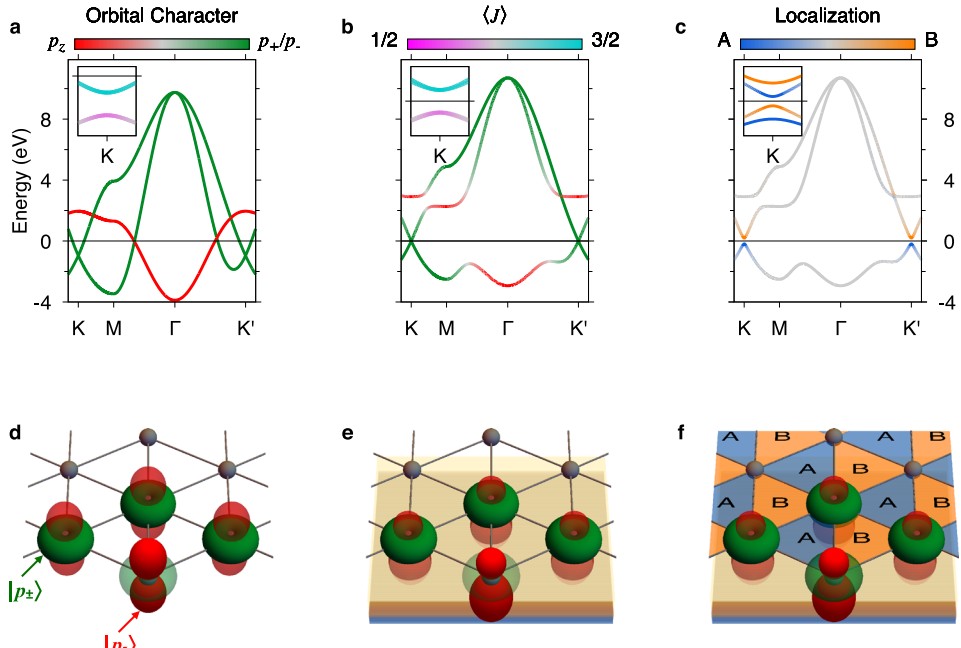

**Fig. 2 Model for a triangular QSHI on a substrate. a–c** Band structures without (with) SOC in the main panels (insets) and corresponding models **d–f** for a $\{p_+, p_-, p_z\}$ basis on a triangular lattice. **a, d** The orthogonality of the $p_z$ and the in-plane orbitals promotes a metallic band structure with Dirac states at $K/K'$ which split in the presence of SOC into states with strong $J = \frac{1}{2}$ and $\frac{3}{2}$ character in valence and conduction band, respectively. **b, e** The presence of a substrate allows for $p_z$ and $p_\pm$ orbital-mixing and opens a hybridization gap. With SOC, a topologically non-trivial insulating ground state is realized (inset to **b**). **c, f** in case of broken inversion symmetry, the blue (A) and orange (B) triangles become inequivalent. Without SOC the system is in the trivial phase and the whole valence (conduction) band localizes in the A- (B-)triangle. The inset to **c** shows the SOC-dominated non-trivial band ordering with spin-split $p_\pm$ states at the valley momenta. The valence (conduction) bands localize now on both triangles of the A/B sublattice. This distinctive A/B localization motif is the same at $K$ and $K'$.

Introducing a honeycomb substrate, such as SiC(0001) (Figs. 1b and 3a), the symmetry is further reduced to $C_{3v}$. As a consequence, ISB renders the A and B halves of the unit cell inequivalent, as schematically illustrated in Fig. 2f. The corresponding non-local term, of strength $\lambda_{ISB}$, acts within the in-plane subspace and opens a gap of size $3\sqrt{3}\lambda_{ISB}$ at $K/K'$. $H^{ISB}$ is diagonal in the spherical harmonics basis and promotes orbital angular momentum (OAM) polarization along the surface normal, competing with the topologically non-trivial local SOC gap (see inset to Fig. 2c). The $p_\pm$ valley-Hamiltonian reads

$$
\begin{aligned}
H(K/K') &= H^{SOC} + H^{ISB}(K/K') \\
&= \lambda_{SOC} L_z \otimes S_z \pm \frac{3\sqrt{3}}{2}\lambda_{ISB} L_z \otimes \mathbb{1}.
\end{aligned}
\tag{2}
$$

Depending on the relative strength of $\lambda_{SOC}$ and $\lambda_{ISB}$, the gap at $K/K'$ is dominated by either of the two types of interaction, which defines the topological phase as indicated in Fig. 1a. All cases shown in the insets of Fig. 2a–c correspond to $\lambda_{SOC} > 3\sqrt{3}\lambda_{ISB}$, i.e., to the topologically non-trivial band ordering. More details on the model can be found in Supplementary Note 2.

As mentioned above, the ISB potential, whose strength depends on the substrate and the bonding distance $d$, distinguishes between A and B (Fig. 1b). Consequently, the charge will tend to localize on the energetically lower triangle. The arrows in Fig. 1c sketch the interference mechanism between the lattice Bloch and the OAM phases determining the charge-density profile (Supplementary Note 2, 5). In the trivial ISB-driven phase, this interplay leads to both spin-valence (conduction) bands at $K$ and $K'$ localizing in the A- (B-)triangle—see band structure without SOC in the main panel of Fig. 2c. The situation changes if the SOC-splitting dominates: the charge associated with the valence band doublet is localized alternatingly in the A and B voids and the

same is true for the two unoccupied eigenvalues, as illustrated by the corresponding colors in the inset to Fig. 2c. A crucial observation is that the charge localization pattern is identical at both valley momenta, since the OAM polarization and the Bloch phase are odd under inversion.

An interference pattern of similar nature has been theoretically discussed in twisted bilayer graphene, though with lattice phases originating from the moiré superstructure, i.e., extending over much longer interatomic distances than in this case[14]. Here, the A/B character represents an emergent honeycomb lattice degree of freedom, intimately linked to the topology of the triangular $p$ model. Its role resembles that of the sublattice index in the graphene Kane–Mele Hamiltonian and it can be associated with the topological gap inversion. Further, it induces the (chirality-dependent) real-space localization of the bulk wave functions that can be measured directly in scanning tunneling microscopy (STM).

**Indenene on SiC(0001).** A monolayer of indium atoms deposited on a Si-terminated 4H-SiC(0001) is an ideal approach to attain a physical realization of our model. Its synthesis is achieved by molecular beam epitaxy, leading to high-quality indenene films as characterized by standard surface science tools (see "Methods" section and Supplementary Note 3). Topographic imaging by STM confirms the well-ordered triangular lattice formed by the In atoms as shown in Fig. 3a. According to the atomic arrangement obtained by total energy minimization within DFT (see "Methods" section) the In atoms bind directly to the uppermost silicon atoms of the SiC substrate (T1 position), as depicted in Fig. 3a, c. This adsorption geometry translates into a $(1 \times 1)$ surface periodicity of the indenene layer with respect to the SiC substrate, with identical in-plane lattice constants as confirmed by the STM line profiles in Fig. 3d[15]. Note in particular the asymmetric height

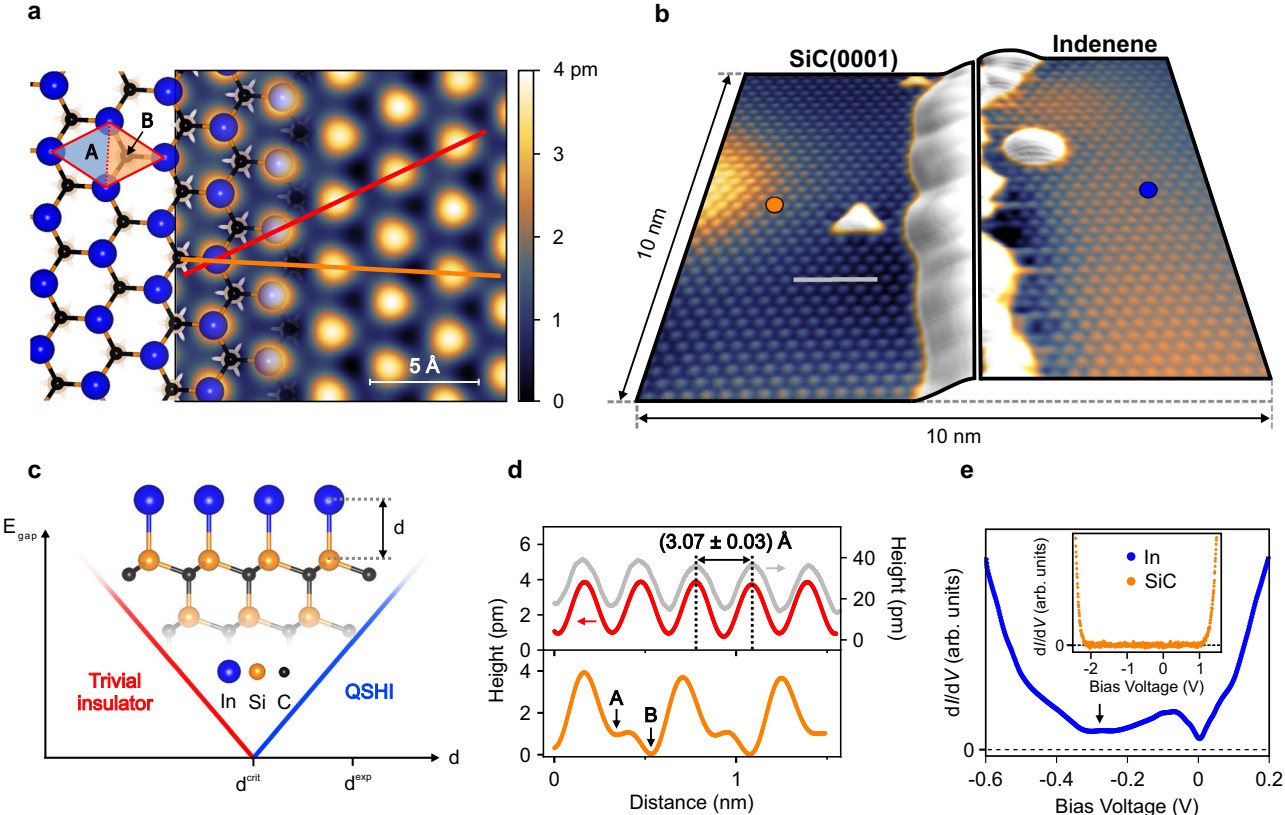

**Fig. 3 Triangular indium monolayer on SiC(0001). a** STM topography image (constant current mode with $V_{set} = 1.5$ V and $I_{set} = 50$ pA) of indenene next to the structural model illustrating the triangular atom lattice forming on the SiC(0001) substrate. Also shown is the (1 × 1) unit cell with its inequivalent A and B halves. At the chosen bias voltage STM probes essentially the out-of-plane In $p_z$ orbitals such that the bright spots directly reflect the positions of the In atoms. **b** Film edge between an indenene monolayer and uncovered SiC. Both lattices appear structurally identical but are distinguishable by their electronic spectra in **e**. Imaging parameters for SiC and indenene are ($V_{set} = 1.65$ V, $I_{set} = 80$ pA) and ($V_{set} = 0.45$ V, $I_{set} = 100$ pA), respectively. **c** Side view of the structural model highlighting the bond length $d$ between the indenene layer and the Si-terminated 4H-SiC substrate. The accompanying graph schematically illustrates the evolution of the energy gap $E_{gap}$ as a function of $d$ in the vicinity of the topological phase transition at $d^{crit}$ (see text for further details). **d** STM height profiles of indenene along the red and orange paths in **a** showing the lattice constant and the asymmetry between the A and B voids of the unit cell, respectively. The gray height profile taken on the uncovered SiC(0001) substrate in **b** proves the identical lattice constants. **e** d$I$/d$V$ spectra measured on an indenene film and SiC substrate. In contrast to the metallic states in the close vicinity of the Fermi level in indenene, SiC exhibits a wide tunneling gap of ~3.2 eV[16]. The black arrow indicates conductance minimum attributed to the Dirac point of indenene.

profile along the orange path (Fig. 3d) which reflects the ISB imposed by the C atoms in the first bilayer of the SiC substrate.

Figure 3b shows an STM image of an indenene layer next to the uncovered SiC substrate. Due to the identical triangular lattice, both surfaces appear structurally indistinguishable. Electronically, though, both systems differ significantly in the differential tunneling conductance d$I$/d$V$ (Fig. 3e), a measure of the local density of states (LDOS). While SiC displays the expected wide energy gap[16], we find for indenene finite spectral weight throughout the entire probed energy region.

Before further analyzing the indenene LDOS, we first turn to its momentum-resolved electronic structure. The red curves in Fig. 4a represent the DFT band structure of the fully relaxed indenene-substrate combination. Apart from the substrate-related bands below −1.5 eV at the center of the Brillouin zone, all other bands are of In $p$ character and reproduce the features seen in our tight-binding model. In particular, we observe a Dirac-like dispersion around $K/K'$ with an additional spin splitting resulting in four distinct bands, as expected in the presence of SOC. Correspondingly, a fundamental band gap of size $E_{gap} = 70$ meV is present (note that the energy scale in Fig. 4 refers to the experimental Fermi level position; DFT per se places $E_F$ in the gap).

Figure 4a also shows the experimental band structure determined by angle-resolved photoelectron spectroscopy (ARPES). It consists of well-defined band features whose dispersions are in remarkable agreement with the DFT prediction. The only notable deviation concerns the position of the Fermi level, which in the experiment is shifted into the upper Dirac half-cone by ≈250 meV due to electronic charge transfer from the strongly $n$-doped substrate (see "Methods" section and Supplementary Note 4 for details). This extrinsic population of the conduction band minimum puts us into a position to probe the bandgap directly by ARPES. For this purpose, Fig. 4a shows a zoom-in of the gap region at the $K$-point. Clearly the quasi-linear dispersions of the upper and lower Dirac cones do not connect to each other (see inset of Fig. 4a). A peak-fit of the energy distribution curves (EDCs) at the $K$-point in Fig. 4c decomposes the spectrum into three distinct peaks, namely the two first valence band states (denoted by VB-1 and VB) and the lowest conduction band state (CB). The next state (CB+1) is essentially cut-off by the Fermi–Dirac function. Extending this decomposition to selected k-vectors around the $K$-point yields the orange markers in Fig. 4b and excellently traces the DFT bands. Their smallest separation is indeed found at $K$, yielding $E_{gap} \approx 125$ meV, in reasonable correspondence with the DFT value, considering that DFT tends to underestimate band gaps.

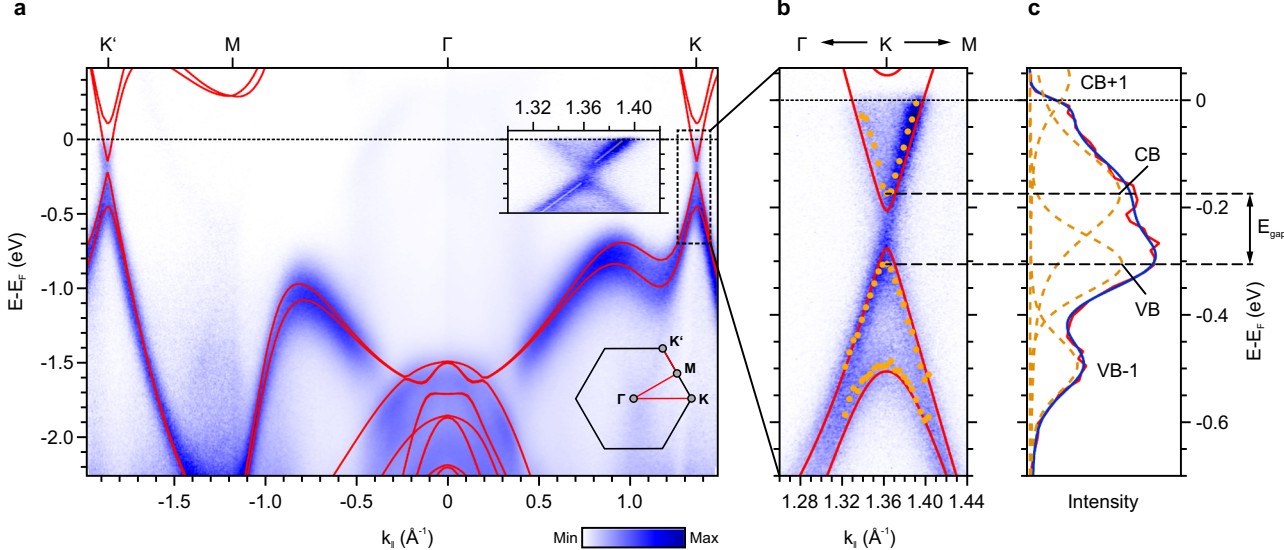

**Fig. 4 Band structure of indenene. a** Comparison of ARPES and DFT band structure (red) and Brillouin zone schematics for the probed high symmetry lines. The faint dispersive features around the Fermi level are artifacts originating from satellite lines of the non-monochromatized He-I radiation (see "Methods" section). The inset shows the dispersive behavior at the $K$-point on a strongly enlarged momentum scale, illustrating the deviation from an ungapped Dirac double-cone. **b** Detailed analysis of the K-point dispersions. The orange markers indicate the peak positions obtained from the peak fitting of the EDCs at selected k-vectors. **c** EDC at the $K$-point (red) and its peak-fit (blue—sum, orange dashed lines—peaks) accounting for the first two valence band states (VB, VB-1) and the lowest conduction band (CB).

With this information at hand, the minimum seen in the experimental LDOS around $-0.3$ eV (arrow in Fig. 3e) is readily identified as the Dirac point, shifted to negative bias voltage due to the finite $n$-doping. The fact that it appears only as a spectral dip rather than a truly vanishing d$I$/d$V$ signal, is a characteristic consequence of the partially occupied CB[17–19] (see also Supplementary Note 5). We note in passing that our STS spectra show nm-scale spatial fluctuations of the chemical potential of the order of $\pm40$ meV (Supplementary Note 5), presumably induced by inhomogeneities in doping concentration as known from related semiconducting substrates[20,21]. In our ARPES data, this effect will be spatially smeared out by the large photon spot (diameter $\approx 1$ mm) and contribute to the energy distribution curve (EDC) peak widths. Noteworthy, the indenene LDOS exhibits a second spectral depression that is always pinned at zero bias, irrespective of local fluctuations. Various mechanisms have been suggested as the origin of such a zero-bias anomaly (ZBA) which, however, depend on the specific probing details[22–24]. It is therefore not considered as an intrinsic feature of the electronic structure.

Having established the existence of a sizable band gap at the valley momenta, we now address the question of its topological character. Our DFT calculation for the fully relaxed structure indicates a non-trivial phase ($\mathbb{Z}_2 = 1$) as derived from the ab initio Wannier charge center movement[25] (Supplementary Note 2). Interestingly, as we have seen from our model, the topology can be tuned by the relative strength of ISB and SOC (Fig. 1a). For our particular case of In/SiC the In-Si bond length $d$ turns out to be the relevant control parameter (Fig. 3c): the smaller the separation to the substrate, the stronger will be the impact of substrate-induced ISB on the indenene layer. For small bond lengths, this implies $\lambda_{\mathrm{ISB}} \gg \lambda_{\mathrm{SOC}}$ (trivial band gap) whereas in the opposite case the system is in the QSHI phase. This picture is confirmed by DFT for fixed (non-relaxed) bond lengths $d$, with the topological transition at $d^{\mathrm{crit}} = 2.57$ Å. The equilibrium bonding distance for our In/SiC system is $d^{\mathrm{DFT}} = 2.68$ Å (see "Methods" section) in excellent agreement with the measured value of $d^{\mathrm{exp}} = (2.67 \pm 0.04)$ Å obtained by X-ray standing wave (XSW) photoemission (Supplementary Note 3). From the distance, we hence get a first, though indirect, hint that indenene is on the non-trivial side of the topological phase diagram. In the following, we present an unambiguous experimental determination of its topology, directly linked to the interference argument anticipated in Fig. 1c.

**Topological classification.** The chiral symmetry of the Dirac states on the triangular lattice can be exploited to access the topological nature directly from 2D bulk properties. Being composed of $p_\pm$ orbitals with defined OAM the valley states assume an $e^{\pm i\phi}$ angular dependence around each atomic center. In combination with the Bloch phase picked up from one lattice site to the next the superposition of neighboring atomic orbitals causes characteristic interference effects, namely the localization of the respective wavefunction at either the A or the B voids of the unit cell, visualized in Fig. 1c. As inferred from our model, the actual information on the trivial vs. inverted character of the bandgap is encrypted in the energy sequence of the A/B localization pattern of the four valley states at K and K' (cf. Fig. 1a and inset of Fig. 2c).

Experimentally, the energy-dependent charge distribution is best addressed by STS, probing the LDOS with atomic resolution. The upper row of Fig. 5a shows d$I$/d$V$ maps taken at selected bias voltages and covering several unit cells. For comparison, the lower row shows the corresponding DFT simulations, accounting for the doping-induced Fermi level shift between experiment and theory. Analogously to the topography map in Fig. 3a, we first calibrate the atomic positions and lattice orientation by probing the indenene $p_z$-dominated states at an experimental bias of 300 mV.

By lowering the tunneling voltage into the energy range of interest and with no other contribution from elsewhere in the Brillouin zone, the STS signal becomes exclusively sensitive to the K/K' valley states. Indeed, at 190 mV the charge maximum has shifted away from the atomic center to the B void of the unit cell. Tuning the bias to smaller and eventually negative values ($-150$ mV) leads to a switch of the charge localization, now

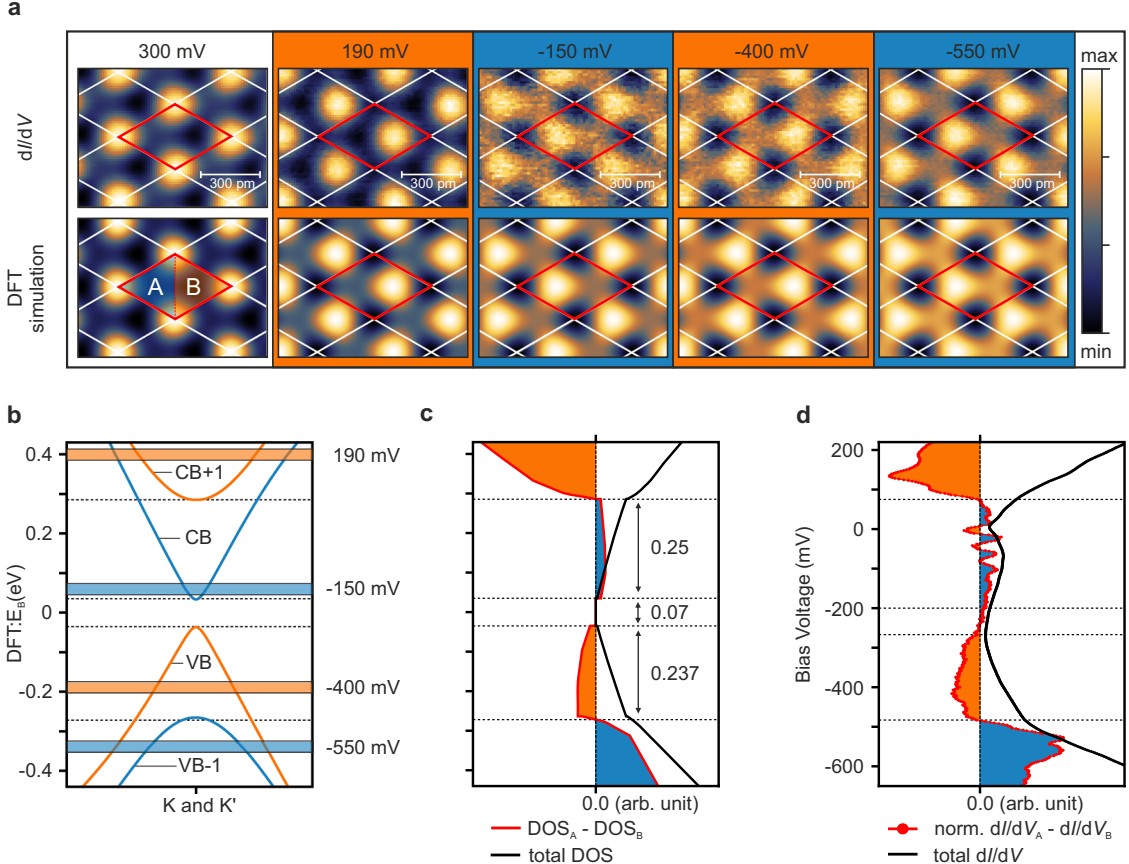

**Fig. 5 Non-trivial Dirac fermion charge localization. a** Spatially resolved constant height d$I$/d$V$ maps taken at different energies in comparison with DFT simulations. The In atom positions are calibrated by mapping the $p_z$ orbital dominated states at 300 mV. Tuning the bias voltage to lower and eventually negative values result in distinct toggling of the LDOS maximum between the two inequivalent A/B parts of the unit cell. **b** Close-up of the DFT band structure and charge localization pattern at both valley momenta. The color code marks the charge maximum at the A (blue) and B (orange) sites. **c** DOS (black line) and A/B difference (color line) from DFT. Blue and orange areas correspond to energies with charge maxima at site A and B, respectively. **d** Differential conductance averaged over the entire unit cell and normalized difference of the d$I$/d$V$ spectra measured at the A/B sites. The noisy behavior near $E_F = 0$ V is attributed to the ZBA. STS data are taken at a constant tip to sample distance (see "Methods" section for more details). Normalized difference spectra and d$I$/d$V$ maps taken at the various tip to sample distances agree qualitatively (see also Supplementary Note 5).

peaking at the A position. Going to even more negative bias voltage repeats the switching pattern, with the charge peak located at B and A at −400 and −550 mV, respectively. Most importantly, the observed alternation of charge localization is in excellent agreement with the corresponding DFT maps of the non-trivial system.

This behavior of the LDOS can be further analyzed by directly comparing the continuous energy dependence of the charge asymmetry between the A and B halves of the unit cell (Fig. 5c, d for DFT and experiment, respectively) to the valley band structure (Fig. 5b). Clearly, the charge difference switches sign each time a new valley state contributes to the LDOS. The noisy behavior of the experimental difference spectrum around the Fermi level is attributed to the ZBA in the total d$I$/d$V$ curve (Fig. 5d) which tends to amplify small extrinsic fluctuations in the A and B charge signals when taking their difference. Overall, we find remarkable qualitative agreement between experiment and theory. Specifically, an alternating ABAB charge localization sequence is established when following the valley states in energy from VB−1 to CB+1, in clear distinction from the AABB sequence predicted for a trivial insulator (Fig. 1a). Our STS data thus confirm that indenene on SiC is a large-gap triangular QSHI.

**Outlook**. From a general perspective, the concept of an emerging honeycomb lattice in the interatomic voids of a triangular atomic

arrangement paves the way for the design of novel 2D QSHIs. Our approach promotes in-plane chiral Dirac fermions at $K/K'$ whose mass term is determined by the interplay of local SOC and ISB. These tunable electronic properties, combined with the simple triangular geometry facilitating large-scale domain growth, are highly desirable for room-temperature transport applications based on the utilization of topologically protected edge states.

The reported topological classification is achieved exclusively by means of local observables and it is intimately linked to the nature of the bulk wave functions. For this reason, it represents an interesting complement to the common identification schemes based on the bulk-boundary correspondence. Its connection to the orbital angular momentum polarization in k-space can be experimentally unveiled by exploiting the coupling of circularly polarized light to the orbital magnetization and Berry curvature[26,27]. It also establishes fast early-stage material screening that is complementary to challenging quantum transport experiments and can become relevant to topology beyond solid-state physics, e.g., in optical lattices of ultra-cold gases[28].

## Methods
**Indenene synthesis, STM, and photoemission measurements**. 4H-SiC(0001) samples (12 mm × 2.5 mm, $n$-type doped (0.01–0.03) Ωcm) with an atomically flat and well-ordered surface were prepared in a gaseous hydrogen dry-etching process[29,30]. Here, 2 slm H$_2$ and 2 slm He both with a purity of 7.0 were

additionally filtered in gas purifiers and eventually introduced in a dedicated ultra-high vacuum (UHV) chamber with a pressure of ~950 mbar. The SiC sample was then etched at 1180 °C for 5 min. The smooth hydrogen passivated SiC sample[29,30] was then transferred in situ to the epitaxy chamber where the surface quality was inspected with low-energy electron diffraction (LEED) prior to the indium epitaxy. After a heating step that removed the H-saturation from the substrate, highly pure indium (99.9999%) was evaporated from a standard Knudsen cell. Excessive indium was reduced thermally until only $(1 \times 1)$ LEED diffraction spots remained (Supplementary Note 3).

STM data were acquired using a commercial Omicron low-temperature LT-STM operated at 4.7 K and a base pressure lower than $5 \cdot 10^{-11}$ mbar. The chemically etched W-tip was conditioned and inspected on an Ag(111) crystal before and after measuring a sample. $dI/dV$ maps were taken at constant height using a standard lock-in technique with a modulation frequency of 971 Hz and modulation voltage of $V_{rms} = 10$ mV. $dI/dV$ curves were recorded using the same lock-in technique. We achieved a semi-constant height mode (CHM) by interrupting the feedback loop at tunneling parameters with featureless topography in constant current mode (CCM) (e.g., at $I_{set} = 50$ pA and $V_{set} = -900$ mV) followed by an approach of the tip to the sample surface by $\Delta z = -2.8$ Å in order to generate a sufficiently large tunneling signal.

ARPES and X-ray photoelectron spectroscopy (XPS) data were recorded in our home-lab photoemission setup from Specs equipped with a hemispherical analyzer (PHOIBOS 100), a He-VUV lamp (UVS 300) generating photons of 21.2 eV, and a 6-axis LHe-cooled manipulator (20 K for ARPES, room temperature for XPS). The base pressure of this UHV setup lies below $1 \cdot 10^{-10}$ mbar. During LHe-cooled measurements the He partial pressure of the differential pumped He-VUV lamp did not exceed $1 \cdot 10^{-9}$ mbar in the UHV chamber.

Room and low-temperature XSW measurements were performed at beamline I09 at Diamond Light Source in UHV environment. The samples were prepared and characterized by ARPES in our home lab before shipping them in situ in a UHV suitcase with base pressure below $1 \cdot 10^{-9}$ mbar. For more details, see Supplementary Note 3.

**DFT calculations**. For our theoretical study of indium on SiC(0001) we employed state-of-the-art first-principles calculations based on the density functional theory as implemented in the Vienna ab initio simulation package (VASP)[31], within the projector-augmented plane-wave (PAW) method[32,33]. For the exchange-correlation potential, the HSE06 functional was used[34], by expanding the Kohn–Sham wave functions into plane-waves up to an energy cut-off of 500 eV. We sampled the Brillouin zone on an $12 \times 12 \times 1$ regular mesh, and when considered, SOC was self-consistently included[35]. The energy decomposed densities are calculated on refined k-grids with a sampling of at least $90 \times 90 \times 1$ and $54 \times 54 \times 1$ for the low-energy states at K and at M, respectively, by selecting all relevant k-points with states inside the investigated energy window with the help of a Wannier Hamiltonian. The indenene low-energy models are extracted by projecting onto In $p$- and SiC $sp^3$-like functions (MLWF) by using the WANNIER90 package[36] to compute the $\mathbb{Z}_2$ topological invariant by following the general method of Soluyanov and Vanderbilt[25]. We consider a $(1 \times 1)$ reconstruction of triangular In on four layers of Si-terminated SiC(0001) with an in-plane lattice constant of 3.07 Å. The equilibrium structure is obtained by relaxing all atoms until all forces converged below 0.005 eV/Å resulting in an In-SiC distance of $d_{In-SiC} = 2.68$ Å. To disentangle the electronic states of both surfaces a vacuum distance of at least 25 Å between periodic replicas in z-direction is assumed and the dangling bonds of the substrate terminated surface are saturated by hydrogen. Structural models are visualized with VESTA[38].

**Tight-binding model**. We consider a triangular lattice with a (In) $p$ basis with a nearest-neighbor interaction given by Slater–Koster parameters[37]. The on-site energies and transfer integrals are extracted from a Wannier Hamiltonian. Detailed information on the model can be found in Supplementary Note 2.

## Data availability
The data that support the plots within this paper and other findings of this study are available from the corresponding author upon reasonable request.

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

## Acknowledgements

We acknowledge Diamond Light Source for time on beamline I09 under proposals NT26419-1 and SI25151-4. The research leading to these results has received funding from the European Union's Horizon 2020 research and innovation program under the Marie Skłodowska-Curie Grant Agreement No. 897276. We gratefully acknowledge the Gauss Centre for Supercomputing e.V. (https://www.gauss-centre.eu) for funding this project by providing computing time on the GCS Supercomputer SuperMUC-NG at Leibniz Super-computing Centre (https://www.lrz.de). We are grateful for funding support from the Deutsche Forschungsgemeinschaft (DFG, German Research Foundation) under Germany's Excellence Strategy through the Würzburg-Dresden Cluster of Excellence on Complexity and Topology in Quantum Matter ct.qmat (EXC 2147, Project ID 390858490) as well as through the Collaborative Research Center SFB 1170 ToCoTronics (Project ID 258499086).

## Author contributions

M.B. and J.E. have realized the epitaxial growth and surface characterization and carried out the ARPES and STM experiments and their analysis. P.E. has conceived the theo-retical ideas and performed the DFT, Wannier and Berryology calculations. On the experimental side, contributions came from P.K.T., J.G., T.-L.L., J.S., S.M., and R.C., while D.D.S. and G.S. gave inputs to the theoretical aspects. R.C. and G.S. supervised this joint project and wrote the manuscript together with all other authors.

## Funding

## Competing interests

The authors declare no competing interests.
