## [Peer Review File · Nature Communications]

REVIEWER COMMENTS

Reviewer #1 (Remarks to the Author):

Large-gap quantum spin Hall insulators are promising candidates for potential room-temperature applications based on Dirac fermions. In this work, the authors prepared and characterized In monolayer (indenene) on SiC by combining STM, ARPES and DFT calculations, and identify the quantum spin Hall phase of this triangular lattice and unveil how a hidden honeycomb connectivity emerges from interference patterns in Bloch wave functions. The experimental results are interesting and important. The theoretical calculations and interpretation seem reasonable. The manuscript is well written. However, there still are some places that are not clear, and clarifications and revisions are needed, which are listed below. If the authors could reasonably address these questions and comments, I would support the publication of this manuscript.

1. A large scale STM image away from the edges is needed to confirm the high quality of indenene, since either XPS or LEED does not reveal the local structure of indenene.
2. The STM image with the edge is splitted (Figure 3d) and doesn't clearly show the edge structure. It is better to add or replace Figure 3d with a complete STM image including SiC, indenene and the edge. A further question, what is the typical edge configuration of In monolayer islands?
3. In addition to the line profiles of indenene (Figure 3c), the line profiles of SiC(0001) should be included, since the claim is that the lattice constant of indenene is identical to SiC(0001).
4. Given the band gap of In monolayer is ~ 125 mV from the ARPES measurements, the quasi-particle band gap measured from STS should be larger than that, why no gap, instead only a dip, is shown in the STS?
5. Because the main focus of this paper is about indenene and the experimental measurements on the band gap of SiC have been reported previously, for Figure 3e, I suggest to change the inset to a main figure with both the dI/dV spectra of indenene and SiC as main figures, or exchange the main figure and the inset.
6. In the current order, Figure 1c is very confusing since no any experimental information about the STM image is described. I suggest to change Figure 1c to completely schematic, or put the theoretical calculations after the experimental results.

Reviewer #2 (Remarks to the Author):

In this manuscript, based on theoretical calculations, STM and ARPES measurements, the authors claim that they identify a quantum spin Hall (QSH) phase in the triangular lattice made of In/SiC. This is an interesting work for designing topological state on surface supported system, and it extends the QSH phase from hexagonal lattice made of Bi/SiC to triangular lattice. However, I have several reservations about this manuscript, and I don't think the theoretically proposed QSH phase is supported by the experimental results, as explained below. Therefore, I cannot recommend it for publication in Nature Communications in its present form.

1. Without the quantum transport measurement, the most important experimental evidence to support the existence of QSH phase is the localized topological edge state in bulk topological gap (similar to ref. 10). However, no STM results are provided in this work. Alternatively, the authors compare the bulk LDOS between the theory and experiment (Fig. 5) to prove the nontrivial band topology. Obviously, this method is too weak to support the QSH phase.

2. The bulk topological gap is extracted from the ARPES measurement. Clearly, there is a strong spectra intensity in the author defined gap region (Fig. 4b), but they still use two peaks (CB and VB) to fit the EDC to show the gap (Fig. 4c). Consequently, the appearance of the experimental gap

strongly depends on the fitting process. Moreover, if this is a global gap, it should induce a small gap region in the STM spectra. However, the inset in Fig. 3e shows that the minimal dI/dV is at the zero energy (not at the arrow position compared to the ARPES). These features make it questionable to claim the existence of the bulk topological gap.

3. In the theoretical model, the authors report two mechanisms to break the band degeneracy. One induces trivial gap, and the other induces nontrivial gap. The bond length between In and Si atom is used to characterize the boundary between trivial insulator and QSH phase in Fig. 3b. Besides the band inversion, I think the small bond length variation induced band structure change will be very small, if the trivial phase can also reproduce the observed ARPES and STM results, the whole story will be different.

Reviewer #3 (Remarks to the Author):

The authors give theoretical and experimental evidence for a new realization of a quantum spin Hall (QSH) insulator in "indenene," a monolayer of indium on a SiC substrate. The theoretical evidence consists of an ab initio calculation and a tight-binding model that matches the low-energy bands of the ab initio calculation. Interestingly, the nontrivial topology arises from a slightly different mechanism than usual: the p_x/p_y and p_z bands overlap causing a nontrivial band inversion; then out of plane mirror-symmetry breaking and spin-orbit coupling together gap all the bands at the Fermi level. (This is in contrast to, for example, graphene, where all bands come from the p_z orbitals, and bismuthene, where all bands come from p_x/p_y orbitals.)

The experimental evidence consists of STM data (showing the symmetry and density of states) and ARPES data that matches the ab initio calculation, up to an overall energy shift. The experimental data does not directly show evidence of the QSH state, i.e., there is no evidence of quantized transport or even gapless edge states. The realization of the QSH state is deduced from matching the experimental bands to the theoretical model and computing the nontrivial topological invariant of the model. Nonetheless, combined, the theoretical and experimental components of the manuscript tell a convincing story of a new platform for the QSH effect, which is exciting and may give rise to improved materials in the future. Thus, I recommend the paper for publication in Nature Communications.

The manuscript is very clearly written. My biggest criticism is that the existence of an experimental band gap is not as clear as the authors make it out to be. If I look at only the experimental data in Fig 4a at K, it would not be clear to me that there is a band gap. It is only after the theoretical peak fitting in Fig 4b that one might believe there is a gap. Can the authors comment more on why the band gap is not more clearly observed in the experiment? Is it temperature, disorder, or something else?

Another question that I have is: what is the upper cut-off of the bond-length d between the indenene layer and the substrate? It seems like as $d \rightarrow$ infinity, the monolayer will either become metallic (no mirror-symmetry breaking), unstable, or both. Could the authors comment on what happens as d increases?

Design and realization of topological Dirac fermions on a triangular lattice
- Response to Reviewers -
(Dated: June 23, 2021)

Reviewers' Comments to the Authors:

Reviewer #1:

Large-gap quantum spin Hall insulators are promising candidates for potential room-temperature applications based on Dirac fermions. In this work, the authors prepared and characterized In monolayer (indenene) on SiC by combining STM, ARPES and DFT calculations, and identify the quantum spin Hall phase of this triangular lattice and unveil how a hidden honeycomb connectivity emerges from interference patterns in Bloch wave functions. The experimental results are interesting and important. The theoretical calculations and interpretation seem reasonable. The manuscript is well written. However, there still are some places that are not clear, and clarifications and revisions are needed, which are listed below. If the authors could reasonably address these questions and comments, I would support the publication of this manuscript.

Response: We thank the reviewer for appreciating our research and for her/his valuable comments. The following paragraph addresses the issues raised and describes their implementation in the new version of our manuscript.

Comment 1: *A large scale STM image away from the edges is needed to confirm the high quality of indenene, since either XPS or LEED does not reveal the local structure of indenene.*

Response: We thank the referee for her/his helpful comment. We have accordingly added overview STM topography scans to the Supplemental Information (Fig. S7) that confirm the high growth quality on different length scales.

Comment 2: *The STM image with the edge is splitted (Figure 3d) and doesn't clearly show the edge structure. It is better to add or replace Figure 3d with a complete STM image including SiC, indenene and the edge. A further question, what is the typical edge configuration of In monolayer islands? (Note from the authors: Fig. 3d changes to Fig. 3b in the revised manuscript)*

Response: We appreciate the referee's remark. However, it has to be pointed out that due their widely different energy gaps the indenene monolayer and the SiC substrate cannot be imaged with atomic resolution using the identical STM set-point (i.e., bias voltage and tunneling current). Instead, different set-points are needed for monolayer and substrate, resulting in the split image of Fig. 3b. The set-point effect is illustrated in Fig. S8 of the Supplemental Information.

As for the edges of the indenene film itself, we wish to add here that they typically consist of flat and zigzag terminations, alternating on short nanometer-sized length scales. Their detailed scrutiny will, however, be topic of a separate systematic study, as they are not in the focus of the present paper. Here we rather analyze the *bulk* properties of indenene which already allow a clean and unambiguous topological classification (see also response below).

Comment 3: *In addition to the line profiles of indenene (Figure 3c), the line profiles of SiC(0001) should be included, since the claim is that the lattice constant of indenene is identical to SiC(0001). (Note from the authors: Fig. 3c changes to Fig. 3d in the revised manuscript)*

Response: We thank the referee for her/his suggestion on this and included a line profile taken on the SiC(0001) film in Figure 3b (gray) which is displayed on top of the indenene line profile (red) in Figure 3d. It clearly demonstrates the identical lattice constants of substrate and monolayer. We adapted the caption and the main text accordingly. Note, that the same information is also contained in Fig. S8c in the Supplemental Information.

Comment 4: *Given the band gap of In monolayer is ~ 125 mV from the ARPES measurements, the quasi-particle band gap measured from STS should be larger than that, why no gap, instead only a dip, is shown in the STS?*

Response: This is indeed a relevant question. As already indicated in the original manuscript, we attribute the finite differential tunneling conductance in the gap region (arrow in Fig. R1a on page 8) to the strong substrate-induced *n*-doping of the In monolayer, i.e., in the occupation of low-lying conduction band (CB) states. The resulting non-zero dI/dV signal in the actual bandgap is a generic effect in such a situation and has been reported for many other degenerately doped semiconductors, e.g., *n*- and *p*-type GaAs(110) [1], *n*-type InSb(110) [2], *n*-type ScN(001) [3] and *n*- and *p*-type H-terminated Si(100) [4, 5]. In order to allow the reader a better comprehension of the underlying mechanism we added a model for the tunneling current at different doping situations to the Supplemental Information (Chapter V.A).

The model was initially suggested by Feenstra *et al.* [1] for GaAs(110) which we adapt here to indenene by changing the gap value to 125 meV and assuming equal band masses for CB and valence band (VB). $I(V)$ and $dI/dV(V)$ curves are calculated for different shifts $E_S = \text{CBM} - E_F$ of the semiconductor Fermi level E_F with respect to the CBM. The results are depicted in Fig. R1b. Panel c and the blue curve represent a situation where E_F is located inside the band gap. No tunneling current flows at bias voltages V in the gap and thus dI/dV is zero. However, if E_F is shifted into the CB, as illustrated in Fig. R1d, a doping-induced current I_D tunnels into the tip, even though V is positioned inside the gap. Since the tunneling barrier depends on the applied bias voltage [1, 6], variations in V also change the barrier height "seen" by the occupied CB states that inevitably result in a finite differential conductance (gray curve). This offset increases for higher CB occupation as in the case of the red curve ($E_S = 200$ meV), representing the Fermi level position observed in ARPES. The doping-induced contribution to dI/dV is shown for lower bias voltages (dashed line). Clearly, the VB contribution exceeds the doping-induced fraction quite rapidly, allowing to neglect the former at lower bias voltages. Please note that the additional current I_D together with the applied modulation voltage complicate the extraction of a quantitative gap value, however, the described mechanism still leaves a clear qualitative signature of the gap.

Comment 5: *Because the main focus of this paper is about indenene and the experimental measurements on the band gap of SiC have been reported previously, for Figure 3e, I suggest to change the inset to a main figure with both the dI/dV spectra of indenene and SiC as main figures, or exchange the main figure and the inset.*

Response: We have followed the reviewer's suggestion and reversed the order of spectra in Figure 3e. The tunneling spectrum from SiC is now the new inset.

Comment 6: *In the current order, Figure 1c is very confusing since no any experimental information about the STM image is described. I suggest to change Figure 1c to completely schematic, or put the theoretical calculations after the experimental results.*

Response: We thank the referee for the suggestion. In order to allow for a clearer introduction to the manuscript, we have replaced the STM data in Figure 1c by a schematic charge density. The caption to Figure 1c has been revised accordingly.

Reviewer #2:

In this manuscript, based on theoretical calculations, STM and ARPES measurements, the authors claim that they identify a quantum spin Hall (QSH) phase in the triangular lattice made of In/SiC. This is an interesting work for designing topological state on surface supported system, and it extends the QSH phase from hexagonal lattice made of Bi/SiC to triangular lattice. However, I have several reservations about this manuscript, and I don't think the theoretically proposed QSH phase is supported by the experimental results, as explained below. Therefore, I cannot recommend it for publication in Nature Communications in its present form.

Response: We appreciate that the reviewer finds our work of interest. Following her/his suggestions we implemented changes to the manuscript in order to strengthen the points raised in the report.

Comment 1: *Without the quantum transport measurement, the most important experimental evidence to support the existence of QSH phase is the localized topological edge state in bulk topological gap (similar to ref. 10). However, no STM results are provided in this work. Alternatively, the authors compare the bulk LDOS between the theory and experiment (Fig. 5) to prove the nontrivial band topology. Obviously, this method is too weak to support the QSH phase.*

Response: As the referee says, one of the most commonly used methods to determine the QSH nature of a material is the experimental detection of its edge modes. Yet, if quantum edge transport is not (yet) possible, it is highly challenging to unambiguously prove the non-trivial origin of such metallic edge states, notwithstanding the power of the STM tool. For example, even in the case of bismuthene (quoted by the referee and Ref. 10 in our manuscript) its identification as QSH insulator is incomplete: while the experiment has established the existence of metallic edge states, their spin-momentum locking - the actual signature of their topological character - is still awaiting experimental confirmation.

Topology is however ultimately encoded in the *bulk* wave functions and indenene offers the rare opportunity to define an unambiguous local observable which is able to directly probe the *bulk* band inversion. Here we hence follow this more fundamental approach which not only allows for an edge-independent topological classification but it also unveils the key interactions and symmetries responsible for the topological band ordering in the material. More precisely, the topology of indenene on SiC manifests itself in a precise pattern of the charge localization of the valley Bloch states. The measured LDOS spectrum indicates the non-trivial phase and categorically differs from the charge localization of the trivial phase (see manuscript Fig. 5 and supplement Fig. S16), i.e. the spatial pattern of the valley LDOS represents a clear-cut indicator of the QSH phase. Addressing the referee's comment on the strength/weakness of this bulk indicator, let us be extremely clear on this essential point: such an unmediated access to the symmetry and localization properties of the bulk wave functions is among the *strongest* supports to the existence of a QSH phase that one can find. Stimulated by the referee's comment, we want to make this argument clearer in the manuscript: In order to stress that using a bulk indicator sensitive to the topology represents an indisputable protocol for the topological classification of materials, we have revised parts of the introduction and the discussion of Fig. 5.

Comment 2: *The bulk topological gap is extracted from the ARPES measurement. Clearly, there is a strong spectra intensity in the author defined gap region (Fig. 4b), but they still use two peaks (CB and VB) to fit the EDC to show the gap (Fig. 4c). Consequently, the appearance of the experimental gap strongly depends on the fitting process.*

Response: We thank the referee for his comments on the experimental identification of the band gap. Concerning the ARPES data, one has to distinguish between the *qualitative* and the *quantitative* analysis of the spectra, as we have done in the manuscript. As for the former, we know the detailed band situation around the K-point very well from the DFT calculations and have been able to experimentally identify all four peaks (CB+1, CB, VB, VB-1) expected for the Rashba split band edges. In particular, we can clearly see that the peaks representing the two gap-defining states (CB and VB) stay always clearly separated in the energy distribution curves at and around the K-point (see also Fig. S13 in the Supplemental Information). In ARPES this is an unambiguous signal that the gap does *not* close. The same information is also conveyed in the inset of Fig. 4a of the manuscript, showing that the extrapolated band dispersion of the valence band does not merge into the conduction band (and vice versa). This indicates massive Dirac fermions which gap the band structure at the K point.

The referee is completely right that for a *quantitative* determination of the gap size we have to resort to peak fitting, as the large and overlapping peak widths at K do not allow a direct analysis. However, line shapes and widths can be well validated away from the K-point (see again Fig. S13), resulting in numerically rather stable fits. In fact, these fit results have been reproduced independently on at least five different samples.

Any remaining uncertainty originates in the large peak widths. These are predominantly caused by the fact that the ARPES signal spatially averages over the local Fermi level fluctuations, which are induced by the high substrate

doping and are of substantial magnitude (as mentioned in the paper and illustrated in more detail in Sec. V.B of the Supplemental Information). In principle, these fluctuations could be suppressed by using undoped SiC, however, in this situation ARPES can no longer be used to measure the gap as the conduction band will be completely depopulated. We anticipate that substantial charging problems in the ARPES spectra may be another unwanted obstacle when using undoped SiC.

Comment 3: *Moreover, if this is a global gap, it should induce a small gap region in the STM spectra. However, the inset in Fig. 3e shows that the minimal dI/dV is at the zero energy (not at the arrow position compared to the ARPES). These features make it questionable to claim the existence of the bulk topological gap.*

Response: This is indeed a relevant question. We want to point out that the small zero bias anomaly, located exactly at the Fermi level, is an extrinsic phenomenon that often occurs in tunneling experiments [7-9] and completely unrelated to the band gap. We already know from ARPES that the gap is actually located ≈ 200 meV below the Fermi level where we indeed observe another dip in the STS signal though with finite amplitude (see Fig. R1a on page 8). As already indicated in the original manuscript, we attribute the finite differential tunneling conductance in the gap region to the strong substrate-induced n -doping of the In monolayer, i.e., to the occupation of low-lying CB states. The resulting non-zero dI/dV signal in the actual bandgap is a generic effect in such a situation and has been reported for many other degenerately doped semiconductors, e.g., n - and p -type GaAs(110) [1], n -type InSb(110) [2], n -type ScN(001) [3] and n - and p -type H-terminated Si(100) [4, 5]. In order to allow the reader a better comprehension of the underlying mechanism we added a model for the tunneling current at different doping situations to the Supplemental Information (Chapter V.A).

The model was initially suggested by Feenstra *et al.* [1] for GaAs(110) which we adapt here to indenene by changing the gap value to 125 meV and assuming equal band masses for CB and VB. $I(V)$ and $dI/dV(V)$ curves are calculated for different shifts $E_S = \text{CBM} - E_F$ of the semiconductor Fermi level E_F with respect to the CBM. The results are depicted in Fig. R1b. Panel c and the blue curve represent a situation where E_F is located inside the band gap. No tunneling current flows at bias voltages V in the gap and thus dI/dV is zero. However, if E_F is shifted into the CB, as illustrated in Fig. R1d, a doping-induced current I_D tunnels into the tip, even though V is positioned inside the gap. Since the tunneling barrier depends on the applied bias voltage [1, 6], variations in V also change the barrier height "seen" by the occupied CB states that inevitably result in a finite differential conductance (gray curve). This offset increases for higher CB occupation as in the case of the red curve ($E_S = 200$ meV), representing the Fermi level position observed in ARPES. The doping-induced contribution to dI/dV is shown for lower bias voltages (dashed line). Clearly, the VB contribution exceeds the doping-induced fraction quite rapidly, allowing to neglect the former at lower bias voltages. Please note that the additional current I_D together with the applied modulation voltage complicate the extraction of a quantitative gap value, however, the described mechanism still leaves a clear qualitative signature of the gap.

Comment 4: *In the theoretical model, the authors report two mechanisms to break the band degeneracy. One induces trivial gap, and the other induces nontrivial gap. The bond length between In and Si atom is used to characterize the boundary between trivial insulator and QSH phase in Fig. 3b (Note from the authors: Fig. 3b changes to Fig. 3c in the revised manuscript). Besides the band inversion, I think the small bond length variation induced band structure change will be very small, if the trivial phase can also reproduce the observed ARPES and STM results, the whole story will be different.*

Response: The referee addresses here an unavoidable challenge when classifying potential topological insulators close to phase transitions. The trivial and non-trivial phase display indeed qualitatively similar band structures as their topological difference is solely encoded in the wave function symmetry. Hence, experimental information on the band structure alone (as measured by, e.g., ARPES or STM/STS) *cannot* be used to discriminate between both phases. However, it should be kept in mind that the spectral function probed by these techniques (specifically, their k - and r -dependent intensities, respectively) carries also valuable information on the wave function. In our manuscript, the fundamental change in the wave function symmetry (see also our response to comment 1) is put forward as an undisputable indicator, as the charge localization has to necessarily change at the phase transition/gap reopening. We hence use the LDOS spectrum as unambiguous marker of the topological phase. From a general perspective, bringing evidence of the wave function's symmetries corroborates the significance of our approach, as this would also hold for narrow gap systems.

The agreement of the experimental and theoretical bonding distance confirms our structural model as our theory predicts a strong dependence of the gap size on the bonding distance. Yet, as correctly noted by the referee, the comparison of the bonding distances *alone* would obviously be too weak to support the QSHI phase. This is precisely the reason why for the topological classification we rely in this study on the robust features of the charge localization

pattern. To avoid the impression, the topological phase is instead derived from the bonding distance, we have slightly revised the wording of the last paragraph of section "*Indene on SiC(0001)*".

Reviewer #3:

The authors give theoretical and experimental evidence for a new realization of a quantum spin Hall (QSH) insulator in "indenene," a monolayer of indium on a SiC substrate. The theoretical evidence consists of an *ab initio* calculation and a tight-binding model that matches the low-energy bands of the *ab initio* calculation. Interestingly, the nontrivial topology arises from a slightly different mechanism than usual: the px/py and pz bands overlap causing a nontrivial band inversion; then out of plane mirror-symmetry breaking and spin-orbit coupling together gap all the bands at the Fermi level. (This is in contrast to, for example, graphene, where all bands come from the pz orbitals, and bismuthene, where all bands come from px/py orbitals.)

The experimental evidence consists of STM data (showing the symmetry and density of states) and ARPES data that matches the *ab initio* calculation, up to an overall energy shift. The experimental data does not directly show evidence of the QSH state, i.e., there is no evidence of quantized transport or even gapless edge states. The realization of the QSH state is deduced from matching the experimental bands to the theoretical model and computing the nontrivial topological invariant of the model. Nonetheless, combined, the theoretical and experimental components of the manuscript tell a convincing story of a new platform for the QSH effect, which is exciting and may give rise to improved materials in the future. Thus, I recommend the paper for publication in *Nature Communications*.

Response: We are grateful for the positive evaluation of our work and for recommending the manuscript for publication. The reviewer raises interesting questions which we carefully address in the following paragraphs and with appropriate changes to the manuscript.

Comment 1: *The manuscript is very clearly written. My biggest criticism is that the existence of an experimental band gap is not as clear as the authors make it out to be. If I look at only the experimental data in Fig 4a at K, it would not be clear to me that there is a band gap. It is only after the theoretical peak fitting in Fig 4b that one might believe there is a gap. Can the authors comment more on why the band gap is not more clearly observed in the experiment? Is it temperature, disorder, or something else?*

Response: We thank the referee for his comments on the experimental identification of the band gap from the ARPES spectra. Here, one has to distinguish between the *qualitative* and the *quantitative* analysis of the spectra, as we have done in the manuscript. As for the former, we know the detailed band situation around the K-point very well from the DFT calculations and have been able to experimentally identify all four peaks (CB+1,CB,VB,VB-1) expected for the Rashba split band edges. In particular, we can clearly see that the peaks representing the two gap-defining states (CB and VB) stay always clearly separated in the energy distribution curves at and around the K-point (see also Fig. S13 in the Supplemental Information). In ARPES this is an unambiguous signal that the gap does *not* close. The same information is also conveyed in the inset of Fig. 4a of the manuscript, showing that the extrapolated band dispersion of the valence band does not merge into the conduction band (and vice versa). This indicates massive Dirac fermions which gap the band structure at the K point.

The referee is completely right that for a *quantitative* determination of the gap size we have to resort to peak fitting, as the large and overlapping peak widths at K do not allow a direct analysis. However, line shapes and widths can be well validated away from the K-point (see again Fig. S13), resulting in numerically rather stable fits. In fact, these fit results have been reproduced independently on at least five different samples.

Any remaining uncertainty originates in the large peak widths. These are predominantly caused by the fact that the ARPES signal spatially averages over the local Fermi level fluctuations, which are induced by the high substrate doping and are of substantial magnitude (as mentioned in the paper and illustrated in more detail in Sec. V.B of the Supplemental Information). In principle, these fluctuations could be suppressed by using undoped SiC, however, in this situation ARPES can no longer be used to measure the gap as the conduction band will be completely depopulated. We anticipate that substantial charging problems in the ARPES spectra may be another unwanted obstacle when using undoped SiC.

Lastly, we can safely exclude disorder as major source of the peak broadening, given the highly ordered indenene lattice seen in our structural characterizations by LEED and STM (see, e.g., Fig. S7 in Supplemental Information). Also the temperature broadening of the bands measured at 20 K should have only a negligible effect.

Comment 2: *Another question that I have is: what is the upper cut-off of the bond-length d between the indenene layer and the substrate? It seems like as $d \rightarrow \infty$, the monolayer will either become metallic (no mirror-symmetry breaking), unstable, or both. Could the authors comment on what happens as d increases?*

Response: We thank the reviewer for raising this important question. Fig. 3c is meant, as a matter of fact, to represent the situation close to the equilibrium bonding distance only. If we push this all the way up to the freestanding limit, we would enter at unrealistically large distances different phases of the indium monolayer, not discussed in our

manuscript. For a perfectly freestanding indenene layer, DFT predicts indeed a metallic band structure comparable to Fig. 2a.

The theoretical study within DFT of a continuously increasing bonding distance unveils two major effects: 1. As expected, one finds a reduction of inversion symmetry breaking (splitting at the valley momenta). 2. The mirror symmetry breaking, which affects the in-plane and p_z hybridization and the p_z on-site energy, also becomes weaker. As illustrated in Fig. 2a and b, the system becomes metallic as the valley Dirac fermions are no longer pinned to the Fermi energy. To conclude, the presented QSHI model relies on a strong substrate-induced mirror symmetry breaking, which defines an upper bound for the bonding distance. We have changed the description of the phase diagram in the caption of Fig. 3c in order to explicitly mention this important aspect: ” *The accompanying graph schematically illustrates the evolution of the energy gap E_{gap} as a function of d in the vicinity of the topological phase transition at d^{crit} (see text for further details).*”

FIG. R1. **a**, Tunneling current $I(V)$ (top) and differential conductance $dI/dV(V)$ (bottom) taken on indenene. **b**, Model for the tunneling current adapted from Feenstra *et al.* [1] showing $I(V)$ (top) and $dI/dV(V)$ (bottom) spectra for different shifts $E_S = \text{CBM} - E_F$. We change the gap value to 125 meV and assume equal masses for CB and VB. Band bending is neglected for simplicity. **c**, Schematic tunneling diagram of a semiconductor with the sample Fermi level E_F positioned in the gap. In the depicted situation the applied bias voltage $eV = E_{F,\text{tip}} - E_F$ is located in the gap and thus no tunneling current is able to flow. $E_{F,\text{tip}}$ is the Fermi level of the tip and VBM the valence band maximum. **d**, Tunneling diagram for a degenerately n -doped semiconductor with the Fermi level above the CBM. For bias voltages set in the gap the occupied CB states contribute the doping-induced tunneling current I_D .

[1] Feenstra, R. M., and Stroscio, J. A., Tunneling spectroscopy of the GaAs(110) surface. *J. Vac. Sci. Technol. B* **5**, 923-929 (1987).

[2] Whitman, L. J., Stroscio, J. A., Dragoset, R. A., and Celotta, R. J, Scanning-tunneling-microscopy study of InSb(110), *Phys. Rev. B* **42**, 7288-7291 (1990).

[3] Al-Britthen, H. A., Smith, A. R., and Gall, D., Surface and bulk electronic structure of ScN(001) investigated by scanning tunneling microscopy/spectroscopy and optical absorption spectroscopy, *Phys. Rev. B* **70**, 045303 (2004).

[4] Pitters, J. L., Piva, P. G., and Wolkow, R. A., Dopant depletion in the near surface region of thermally prepared silicon (100) in UHV. *J. Vac. Sci. Technol. B*, **30**(2), 021806 (2012).

[5] Fukutome, H., Takano, K., Yasuda, H., Maehashi, K., Hasegawa, S., and Nakashima, H., Scanning tunneling microscopy study of the hydrogen-terminated n - and p -type Si(001) surfaces. *Appl. Surf. Sci.*, **130-132**, 346-351 (1998).

[6] Feenstra, R., Stroscio, J. A. Fein, A. Tunneling spectroscopy of the Si(111) 2×1 surface. *Surf. Sci.* **181**, 295 - 306 (1987).

[7] Ming, F., Smith, T. S., Johnston, S., Snijders, P. C., and Weitering, H. H., Zero-bias anomaly in nanoscale hole-doped Mott insulators on a triangular silicon surface, *Phys. Rev. B* **97**, 075403 (2018).

[8] Zhang, Y., Brar, V. W., Wang, F., Girit, C., Yayan, Y., Panlasigui, M., Zettl, A., and Crommie, M. F., Giant phonon-induced conductance in scanning tunnelling spectroscopy of gate-tunable graphene, *Nature Phys.* **4**, 627-630 (2008).

[9] Butko, V. Y., DiTusa, J. F., and Adams, P. W., Coulomb Gap: How a Metal Film Becomes an Insulator, *Phys. Rev. Lett.* **84**, 1543-1546 (2000).

REVIEWERS' COMMENTS

Reviewer #1 (Remarks to the Author):

The authors have satisfactorily addressed my comments and concerns and significantly improved the quality of the manuscript. I recommend publication of this manuscript in Nature Communications.

Reviewer #2 (Remarks to the Author):

In the authors' response to my report, they have given a detailed explanation, demonstrated the validity of bulk indicator to classify the QSH phase, and identified the existence of bulk topological gap. Overall, the revised manuscript is suitable for publication in Nature Communications.

Reviewer #3 (Remarks to the Author):

The authors have satisfactorily answered my questions, therefore I recommend the paper for publication.